# Urodynamic Parameters and Continence Outcomes in Asymptomatic Patients with Ileal Orthotopic Neobladder: A Systematic Review and Metanalysis

**DOI:** 10.3390/cancers16071253

**Published:** 2024-03-22

**Authors:** Anastasios D. Asimakopoulos, Enrico Finazzi Agrò, Thierry Piechaud, Georgios Gakis, Richard Gaston, Eleonora Rosato

**Affiliations:** 1Unit of Urology, Fondazione PTV Policlinico Tor Vergata, 00133 Rome, Italy; 2Department of Surgical Sciences, Unit of Urology, University of Rome Tor Vergata, 00133 Rome, Italy; finazzi.agro@med.uniroma2.it (E.F.A.); eleonoraros92@gmail.com (E.R.); 3Unit of Urology, Clinique Saint-Augustin, 33074 Bordeaux, France; pthpiechaud@hotmail.fr (T.P.); gastonrich@wanadoo.fr (R.G.); 4University Clinic and Polyclinic of Urology, University Hospital of Halle (Saale), D-06120 Halle, Germany; georgios.gakis@uk-halle.de

**Keywords:** urodynamics, cystectomy, urinary diversion, review, systematic, bladder, daytime urinary incontinence, nighttime urinary incontinence

## Abstract

**Simple Summary:**

Among the various options for urinary diversion following radical cystectomy, the orthotopic neobladder most closely resembles the original bladder both in location and function. However, a significant number of patients with these reservoirs have dysfunctional voiding. Our objective here is to provide the first systematic review focusing on the urodynamic and continence outcomes of ileal orthotopic neobladders. By summarising these important outcomes, the current paper may represent the reference manuscript for outcome comparison in future papers. The manuscript also describes the methodology of the urodynamic evaluation of the neobladders, highlighting the frequent lack of precise indications, accurate guidelines (at the state of the art, the same parameters used for the native bladder are also used for the ileal neobladders), standardised definitions, and standard values for outcome comparison. By underlining these gaps, our systematic review may aid future studies in having more adequate designs and will allow for a more accurate functional evaluation of the patients harbouring an ileal neobladder.

**Abstract:**

Introduction: The orthotopic neobladder is the type of urinary diversion (UD) that most closely resembles the original bladder. However, in the literature the urodynamic aspects are scarcely analysed. Objective: To provide the first systematic review (SR) on the urodynamic (UDS) outcomes of the ileal orthotopic neobladders (ONB). Continence outcomes are also presented. Methods: A PubMed, Embase and Cochrane CENTRAL search for peer-reviewed studies on ONB published between January 2001–December 2022 was performed according to the Preferred Reporting Items for Systematic Review and Meta-analysis (PRISMA) statement. Results and Conclusion: Fifty-nine manuscripts were eligible for inclusion in this SR. A great heterogeneity of data was encountered. Concerning UDS parameters, the pooled mean was 406.2 mL (95% CI: 378.9–433.4 mL) for maximal (entero)cystometric capacity (MCC) and 21.4 cmH_2_O (95% CI: 17.5–25.4 cmH_2_O) for Pressure ONB at MCC. Postvoid-residual ranged between 4.9 and 101.6 mL. The 12-mo rates of day and night-time continence were 84.2% (95% CI: 78.7–89.1%) and 61.7% (95% CI: 51.9–71.1%), respectively.Despite data heterogeneity, the ileal ONB seems to guarantee UDS parameters that resemble those of the native bladder. Although acceptable rates of daytime continence are reported the issue of high rates of night-time incontinence remains unsolved. Adequately designed prospective trials adopting standardised postoperative care, terminology and methods of outcome evaluation as well as of conduction of the UDS in the setting of ONB are necessary to obtain homogeneous follow-up data and to establish UDS guidelines for this setting.

## 1. Introduction

Radical cystectomy (RC) is the gold standard treatment for organ-confined muscle-invasive bladder cancer (MIBC) and for very high-risk non-muscle-invasive bladder cancer (NMIBC).

Since the early 1900s, surgeons have sought an optimal method to replace the original bladder when it must be removed [1]. The individual selection of urinary diversion (UD) is usually based on the balance between oncological control and quality of life (QoL), taking into account the technical feasibility as well as the health status and life expectancy of the patient. Incontinent urinary diversions are performed much more often than continent ones, particularly in patients with complex medical or surgical histories and/or those that have a history of previous radiotherapy. As a common denominator, these diversions require an external ostomy appliance, and they consequently affect the body image of the patient. On the other hand, the orthotopic neobladder (ONB) is the type of UD that most closely resembles the original bladder, both in location and function [1]. All parts of the small and large intestine as well as the stomach have been studied for the construction of orthotopic reservoirs, but some studies showed advantages for the ileum over any other segment [2].

ONB allows for voluntary voiding, avoids the need for urinary control devices, and only requires self-catheterization in a minority of patients [2,3]. Patients with ONB may present better QoL compared to patients with incontinent UD [2]. However, a significant number of patients with ONB have dysfunctional voiding [3], regardless of the intestinal segment that is used. This voiding dysfunction may affect both the storage and voiding phases, and it may occur during the daytime, nighttime, or both [3]. Between 4% and 25% of patients perform intermittent self-catheterization for incomplete emptying, and many studies showed that failure to empty is more frequent in female patients [2,3]. Urethroneovesical anastomotic stricture is a cause of obstruction after this type of surgery [2].

To obtain better functional outcomes and reduce the incontinence rate in patients with ONB, many surgical aspects are developed, such as using an adequate length of ileum and an ellipsoid or spherical configuration. The nerve-sparing technique and the prevention of injury to the pelvic floor could reduce daytime and nighttime continence [2].

As can be seen from the literature, in many ONB series the functional aspects are less analysed compared to the surgical and oncological outcome, and the prevalence of lower urinary tract symptoms (LUTS) is underestimated [3,4,5]. Moreover, there is no consensus about functional outcome evaluation (continence or invasive urodynamic study), and the European Association of Urology’s (EAU) and American Association of Urology’s (AUA) Guidelines do not report evidence on these aspects [4,5] nor any recommendations.

However, some authors investigated the role of an invasive urodynamic study on the neobladder to objectify the functional outcomes of this type of UD in patients’ series. Urodynamic studies represent the most objective method for the functional assessment of the ONB, although their results depend on the type and configuration of the ONB as well as the time interval from surgery [3]. Currently, the same UDS parameters are applied in an ileal neobladder as in an intact bladder without considering that the bowel was not originally evolved to store or void urine. Furthermore, in the literature, there is limited evidence concerning the timing as well as good practice criteria, terminology, and parameters to properly describe the filling and voiding phases of the ONB.

The primary aim of this systematic review (SR) is to analyse and summarise the urodynamic parameters of the “normal” ileal ONBs (i.e., UDS performed for functional evaluation of the neobladders and not for ONB patients being assessed for abnormal voiding). The secondary aims are to report continence outcomes as well as the main technical characteristics of the ileal neobladders.

## 2. Methods

The present SR and meta-analysis were performed and reported according to the Preferred Reporting Items for Systematic Reviews and Meta-analyses (PRISMA) statement [6]. PRISMA Checklist and A MeaSurement Tool to Assess systematic Reviews (AMSTAR) Checklist were completed (Appendix A). The study has not been registered.

### 2.1. Information Sources, Search Strategy and Selection Process

We performed a comprehensive literature search on Pubmed, Embase and Cochrane CENTRAL including peer-reviewed studies published between January 2001 and December 2022. The keyword search was performed using both Medical Subject Headings (MeSH) terms and free text including (“Neobladder” OR “Orthotopic reconstruction” OR “Bladder substitution” OR “Orthotopic urinary diversion”) AND (“Urodynamics”).

### 2.2. Inclusion and Exclusion Criteria

Two authors (ADA and ER) independently screened all titles, abstracts and full-text records against the eligibility criteria by collecting them in an Excel sheet after discussing and resolving any divergence. No automation tools are used.

For this SR, we included cohort and case-control studies as well as randomised trials. All included studies were on patients having UDS for routine ONB function; thus, no study included ONB patients being evaluated for abnormal voiding. Manuscripts reporting <5 cases of ONB or not reporting UDS data were excluded, as were review articles, meta-analyses, surveys, expert opinions, and editorials. Abstracts not followed by a full-text manuscript were excluded unless data could be retrieved from the abstract for analysis. Only English written studies were included in this systematic review.

Studies referring to a paediatric cohort, as well as studies assessing non-ileal ONB (gastric, sigmoid, or ileocolic/colic), reporting on cystectomies and ONB performed for indications different from bladder cancer (i.e., locally advanced rectal cancer infiltrating the bladder), as well as prostate-sparing cystectomies, were excluded.

### 2.3. Data Collection Process and Data Items

Eligible outcomes were broadly categorised as follows: (1) UDS data; (2) continence outcomes; and (3) baseline characteristics and technical issues. These outcomes are summarised in Table 1.

Data from studies with a minimum follow-up of 1 month to a maximum of 240 months were eligible. Some trials reported data at multiple follow-up time points (1–3–6 months, etc.); to combine data for synthesis, only 12-month outcomes were considered.

The number of participants in each included study as well as the number of subjects at each follow-up time point, the characteristics of the participants (mean age and sex), the type of study (retrospective versus prospective versus RCT), and the type of neobladder have also been extracted. Regarding the outcomes of interest, for the continuous data (e.g., UDS parameters), means and standard deviations (SD) were extracted. If this information was not available, medians and intervals (range or interquartile range) were extracted and inserted into the Excel sheet.

### 2.4. Effect Measures

The analysis of the rates of daytime and nighttime continence as well as of the means of the UDS parameters was planned. 12-mo day- and night-time continence rates were analysed in terms of proportion over the total number of patients presented at 12-mo follow-up. The pooled proportions and their 95% confidence interval (95% CI) were reported. We also analysed the mean UDS parameters at 12-month follow-up; the pooled means and their 95% confidence interval (95% CI) were reported.

### 2.5. Synthesis Methods

Before undertaking statistical synthesis, for each study, the percentage of dropout was calculated; studies with a dropout of ≥50% were not considered in the analyses. For the UDS parameters, in the absence of the standard deviation (SD), but the range values were reported, the SD was approximately estimated using this formula
$$\mathrm{SD}=\frac{\left(max-min\right)}{4}$$

As the proportion of patients with day- or night-time continence was deemed to be highly variable according to the sample size of each study, a random-effects model was chosen to calculate the overall proportion that could be expected. DerSimonian-Laird random-effects variance estimator was used. The 95% CI around the pooled proportion was reported.

In order to calculate the pooled mean of UDS parameters, a random-effects model was chosen with a DerSimonian-Laird random-effects variance estimator. The 95% CI around the pooled mean was reported.

The extent and impact of between-study heterogeneity were assessed by inspecting the forest plots and quantified by calculating the tau-squared and the I-squared statistics, respectively. The 95% CIs (uncertainty intervals) around the I-squared were reported. The results of the main analyses were represented graphically by the forest plot.

It was impossible to calculate the overall mean of the length of harvested ileum for the ONB configuration since the SD or range was rarely reported. Thus, a weighted mean was calculated, assuming as weight the sample size of the study.

No meta-analysis was performed on the mean PVR due to the extreme variability in the estimates reported by each study; only minimum and maximum values were reported.

To explore the possible causes of variation in results across the studies regarding daytime/nighttime continence and UDS parameters (MCC and P_ONB_ at MCC), subgroup analysis was performed considering the type of ONB (Y-shape, S-shape, W-Hautmann, Camey II and Studer).

The results of subgroup analysis were represented in a table, reporting for each subgroup the number of included studies, the estimated pooled parameters (proportion or mean), the relative 95% CI, the quantification of heterogeneity (I-squared, I2), and the *p* value of the test for subgroup differences (psubgroup).

To assess the robustness of the synthesised results, a sensitivity meta-analysis restricting the analysis to trials that considered only male subjects has been conducted.

A *p* value < 0.05 was considered statistically significant. All of the analyses were run in R 4.2.3 (R Core Team, 2022) and the meta package (v4.17-0; Balduzzi et al., 2019).

### 2.6. Reporting Bias Assessment

To assess small-study effects, funnel plots for meta-analyses including at least 10 trials of varying size were generated. In the funnel plot, the effect estimates are plotted against a measure of precision, usually the standard error (SE) of the effect estimate. The test for asymmetry was applied only if the number of included studies was ≥10. The Peters test was used in the meta-analysis of single proportions (day- and night-time continence). The Egger test was used in the meta-analysis of the mean (UDS parameters). Once the presence of asymmetry in the funnel plot had been detected, sensitivity analyses were conducted to adjust the effect estimate for this bias by applying the trim-and-fill method. The trim-and-fill method first trims studies from the funnel plot until it becomes symmetric; in a second step, it adds mirror images of removed studies to the original funnel plot; finally, it calculates the adjusted effect estimate based on the original and added studies.

## 3. Results

### 3.1. Study Selection

Following an initial search, a total of 143 publications were identified through database searching as potentially eligible articles. Figure 1 provides a diagram of the flow of information through the different phases of this SR according to the PRISMA criteria [6].

Finally, fifty-nine manuscripts were included (Table 2).

### 3.2. Study Characteristics

#### 3.2.1. General Aspects

The vast majority of articles included in our SR were retrospective, with only a few (28.8%) prospective trials [8,12,13,15,21,22,23,25,33,46,48,49,53,60,63,64,65]. Only 13 (22%) report institutional review board approval [8,13,14,15,20,21,22,25,33,40,53,64,65].

The mean and median age were 61.2 years and 62.4 years, respectively (25–75th percentile: 58.7 to 65), with a range from a minimum of 42 years [30] to a maximum of 71.2 years [26]. Thirty—five out of 59 studies (59.3%) included patients of both sexes; 19 (54.3%) reported only males; and 5 (14.3%) studied only female patients. Comorbidities were reported in eight studies [7,8,13,30,37,53,64,65]. Time of follow-up is reported in 44 (74.6%) studies, but in 15 (23.4%) studies, this data is lacking [7,9,19,23,32,36,39,41,43,46,48,51,53,60,61]. Follow-up ranges from a minimum of 5 months [35] to 179.2 months [31], for a mean value of 42.7 months and a median value of 32.9 (25–75th percentile: 19 to 60.5 months).

#### 3.2.2. Surgical and Technical Aspects

Only 29 studies (17 open [12,14,15,16,17,21,22,24,25,27,30,33,35,40,41,60,63] 3 laparoscopic [50,53,61], 5 robotic [8,20,28,32,64], 1 mixed open/laparoscopic technique [54] and 3 mixed open/robotic [13,23,65]) provided a thorough description of the surgical technique and time that were needed both for the extirpative (cystectomy) and reconstructive parts of the operation.

Twenty (33.9%) studies [8,10,14,15,17,23,26,27,32,40,42,44,45,50,51,53,57,61,64,65] reported mean overall operating times. Only 6 studies (10.2%) [23,26,50,51,57,64] specified OT for the reconstructive part of the surgery.

The range of harvested ileum was from a minimum value of 25 cm [25] to a maximum of 70 cm [7]. Seven studies did not provide this data [9,13,20,36,51,54,61]. The distribution of the ileal neobladders based on the technique is graphically depicted in Figure 2.

Technical issues are summarized in Table 3.

#### 3.2.3. Continence and Urodynamic Data

All included studies reported on UDS evaluation, but only 31 of them [7,8,18,19,21,22,23,24,25,28,33,34,35,37,38,39,43,46,48,49,50,51,52,53,55,56,58,59,62,64,65] were compliant with the ICS Good Urodynamics Practices [66]. Twenty-four studies (40.7%) [10,11,12,14,15,16,17,19,21,25,26,27,29,30,33,39,42,44,47,53,54,56,57,64] repeated the UDS evaluation at different time-points during follow-up to evaluate the chronological changes of the ONB.

Only 11 studies [8,11,12,13,18,21,37,38,52,64,65] reported the free uroflowmetry outcomes as performed before the invasive UDS evaluation.

Concerning the filling phase of the UDS, the most reported parameters (39 studies, 66.1%) were MCC and compliance [7,8,12,13,14,15,16,19,20,21,22,23,24,25,29,30,32,33,34,35,37,38,40,41,43,45,46,47,49,51,52,53,56,58,60,62,63,64,65]. Thirteen studies [8,21,29,34,37,45,48,49,50,52,56,59,63] documented the absence of persistent peristaltic contractions during the filling phase of the ONB.

Concerning the voiding phase Qmax, PVR and ONB pressure at MCC were more commonly reported (24 studies, 40.7%) [7,9,14,15,19,21,23,24,25,27,29,34,37,39,40,41,43,44,46,47,52,57,64,65]. The number of patients recurring to CIC is reported in 3 (5.1%) studies [13,37,52].

The timing of continence evaluation and the adopted definitions of continence showed great variability among the included studies. The most common definition was 0-1 pads/days as adopted by 34 studies (55.6%) [8,10,11,12,14,16,19,20,22,23,24,25,27,28,29,30,31,33,34,35,37,39,40,44,45,47,49,52,55,56,59,63,64,65]. In all studies, the information about voiding diaries (number of voidings per day and night) as well as the mean number of pads was lacking.

Twenty-two studies reported the use of validated questionnaires to evaluate QoL, lower urinary tract symptoms (LUTS), and continence. Only nine studies [10,12,13,14,20,35,39,55,58,60,64] indicated which questionnaire was used (King’s Quality of Life, EORTC QLQ-30, IONB-PRO, UDI-6, IIQ-7, FACT—BL score, FACT-BL, ICIQ).

### 3.3. Meta-Analysis of UDS Data

A total of 19 studies were included in the meta-analysis of MCC values at 12 months with a total of 1425 subjects: the estimated pooled mean was 406.2 mL (95% CI: 378.9–433.4 mL), with high heterogeneity (I2 = 99%, 95% CI: 99.4–99.6%; *p* < 0.001; Figure 3A).

A total of 13 studies were involved in the meta-analysis of P_ONB_ at MCC, with a total of 1008 subjects. The estimated pooled mean was equal to 21.4 cmH_2_O (95%CI: 17.5–25.4 cmH_2_O), with high heterogeneity (I2 = 95%, 95% CI: 99.4–99.6%; *p* < 0.001; Figure 3B).

A total of 13 studies reported data about PVR, with a total of 1264 subjects. 12-mo PVR minimum and maximum values were 4.9–101.6 mL.

#### Subgroup Analysis and Sensitivity Analysis for UDS Data

Considering the statistically significant heterogeneity among studies in both MCC and P_ONB_ at MCC meta-analyses, subgroup analysis was performed considering the type of ONB (Appendix B Table A1). The test for subgroup differences indicates that it had a significant effect on the MCC mean estimate (psubgroup < 0.001). Subgroups involve a different number of studies, but subgroups with a larger number of studies have fewer participants. However, there is substantial unexplained heterogeneity between the trials within each of these subgroups (W-Hautmann: I2 = 89%; Studer: I2 = 97%; Y-Shape: I2 = 78%); therefore, the validity of the MCC mean estimate for each subgroup is uncertain (Appendix B Table A1).

Concerning the 12-mo P_ONB_ at MCC, the subgroup analysis shows a significant effect of the type of ONB (psubgroup < 0.001). The subgroups involve a different number of studies and subjects (Appendix B Table A1). Thus, the validity of the P_ONB_ at MCC mean estimate for each subgroup is uncertain.

We also performed a sensitivity analysis considering only studies that enrolled only male subjects. For MCC, the analysis was performed considering 4 studies [19,25,27,42], with a total of 671 total male patients; for P_ONB_ at MCC, the analysis was performed considering 3 studies [19,25,27], with a total of 640 total male patients. The results were no longer different from those obtained with the main analyses (Appendix C Figure A1).

### 3.4. Meta-Analysis of Continence Outcomes

Concerning the 12 mo day and night-time continence, 16 studies (with a dropout rate of <50%) were included in the analysis, with a total of 1671 patients.

A total of 1407 daytime continence events were observed, with an estimated pooled proportion of 84.2% (95% CI: 78.7–89.1%). The heterogeneity was high (I2 = 85%, 95% CI: 76.8–90.1%; *p* < 0.01) (Figure 3C).

A total of 1109 night-time continence events were finally observed, with an estimated pooled proportion of 61.7% (95%CI: 51.9%–71.1%). The heterogeneity was high (I2 = 93%, 95% CI: 89.9–94.9%; *p* < 0.001) (Figure 3D).

#### Subgroup Analysis and Sensitivity Analysis for Continence Data

A subgroup analysis was performed to evaluate the impact of the type of ONB on the heterogeneity in daytime continence. A statistically significant subgroup effect (psubgroup = 0.02) was evidenced. However, more trials (and participants) contributed to the Y-shape subgroup (5 studies and 458 total subjects), to the W-Hautmann subgroup (4 studies and 268 total subjects) than to the Studer subgroup (2 studies and 75 total subjects), and to the Camey II subgroup (1 study and 606 subjects) (Appendix B Table A2).

The type of ONB did not impact the rates of nighttime continence as well (*p*_subgroup_ = 0.69). As for daytime continence, the number of studies included in the subgroup analysis differs for each ONB type (Appendix B Table A2).

For daytime continence, the sensitivity analysis was performed considering 6 studies [14,25,27,29,39,42] with a total of 918 male patients. A total of 771 events and 583 events for daytime and nighttime continence were observed. The overall proportion of patients with daytime continence was very similar to the main results: the pooled proportion was of 84.8% (95% CI: 75.7–92.2%), confirming the main results (Appendix C Figure A2). The overall proportion of patients with night-time continence was 62.9% (95% CI: 51.1–74%), quite higher than the main results (Appendix C Figure A2).

### 3.5. Reporting Biases of Metanalysis

Publication bias was assessed by funnel plots. The random effect model is represented by a dashed line on which the funnel is centered. The funnel plot of the meta-analysis of MCC seemed asymmetrical (Figure 4A), as confirmed by the Egger test (*p* = 0.001). Performing the trim-and-fill method, nine studies were added to the meta-analysis (Figure 4B), leading to an adjusted random effects estimate of the MCC mean equal to 329.8 (95% CI: 286.2–373.35).

The funnel plot of the meta-analysis of P_ONB_ at MCC seemed to not be asymmetrical (Figure 5A), as confirmed by the Egger test (*p* = 0.056).

Figure 5B,C refer to meta-analyses of daytime continence and nighttime continence proportion, respectively. Both seemed to be not asymmetric, as also confirmed by the Peters test (daytime continence: *p* = 0.08 and nighttime continence: *p* = 0.29).

## 4. Discussion

Ileal ONB reconstruction is one of the options for UD after RC for bladder cancer [66,67]. Despite the complexity of both the surgical procedure and the postoperative management, patients with ONB void via the native urethra and they may show better QoL outcomes compared to patients with incontinent UD [66].

Ideally, the ONBs should represent low-pressure reservoirs with adequate capacity to preserve the urinary continence and to protect the upper urinary tract function [66,68,69,70,71]. However, the evaluation of the functional aspects during the follow-up of patients with ONB is often neglected. The current SR demonstrates, in fact, that only few studies used bladder cancer- and neobladder-specific questionnaires [66,69,70] as well as ‘neobladder diaries’ that focus on functional outcomes such as urinary incontinence or QoL after surgery.

Urodynamics could provide objective information on lower urinary tract function and symptoms through the measurement of various volume and pressure parameters. The ICS standard urodynamic test consists of uroflowmetry and PVR, plus transurethral cystometry and pressure-flow studies.

Both non-invasive and invasive UDS data are of outmost importance in the functional evaluation of patients with ONB, representing the only method to objectively assess the function of these neo-reservoirs. UDS in patients with ONB is not a new effort, since multiple studies have been conducted to investigate long-term changes in urodynamic parameters or to compare differences in urodynamic parameters between orthotopic neobladders with various intestinal segments [10,11,12,14,15,16,17,18,19,21,22,23,25,26,27,29,30,33,34,39,42,43,44,47,48,49,52,53,54,55,56,57,64,65]. Unfortunately, the timing from the surgery to the UDS in the neobladder is usually not adequately defined. Some authors suggest that UDS should be performed one year after the creation of the ONB, since some months are necessary to stabilise the ONB and its capacity [35]. In our review, 52.5% [8,10,11,12,15,16,17,19,20,21,23,26,27,29,30,32,33,35,39,42,44,47,48,49,50,53,54,56,57,65] of the included papers had performed UDS less than one year after surgery, with 50% of them repeating UDS at 12 months [10,11,15,16,17,19,22,26,27,42,53,54,57,64]. The absence of good urodynamic practices for ONB and the different time points of UDS evaluation for each paper cause a vast heterogeneity of data. Thus, it seems hard to group and graphically depict the chronological changes of the UDS parameters of the ONBs in basic patterns.

A neobladder filling capacity of 300–500 mL is usually recommended for a mature ONB. The pressure produced inside the reservoir, depending on the size and configuration of the ileal segment, influences day and night continence [69,70]. In this study, the pooled mean MCC was 406.2 mL (95% CI: 378.9–433.4 mL). P_ONB_ at MCC is another important UDS parameter since high-pressure reservoirs could be associated with vesicoureteral reflux (leading to renal function deterioration and kidney impairment over time) and urinary incontinence. In our review, the pooled mean value for this parameter was 21.4 cmH_2_O (95% CI: 17.5–25.4 cmH_2_O). At 12 months, the PVR range was 4.9–101.6 mL.

The presence of (residual) intestinal peristalsis that may impact ONB compliance and continence is another scarcely reported data, with only 17 (27.8%) studies reporting on this [8,20,21,29,34,37,45,48,49,50,52,56,59,63,64,65].

As highlighted by the results of this SR, there is no consensus about the urodynamic assessment of the intestinal neobladder. The same parameters applied to an intact bladder are used for orthotopic neobladders that, being created out of intestinal segments, show innate differences from the original bladder in terms of sensory and motor functions. The presence of many retrospective studies contributes to the incompleteness and heterogeneity of the data. Furthermore, there are many missing parameters in the UDS evaluations of the included papers. For example, only 11 studies [8,11,13,18,21,37,38,52,64,65] reported the free uroflowmetry outcomes as performed before the invasive UDS evaluation.

Moreover, in all papers, both the UDS and continence outcomes are presented globally, with the patients being studied as a unique population without being subdivided by sex. Thirty-five studies include both males and females [7,8,9,10,11,12,15,16,17,20,21,22,30,31,32,33,34,35,36,38,40,41,43,48,50,51,53,55,57,58,61,62,63,64,65], with only nineteen of them [8,10,11,12,15,16,20,21,31,33,35,40,41,48,51,53,57,62,65] reporting the exact number for each sex. Finally, no included study provided any evaluation of the correlation between MCC, nocturnal polyuria, and nighttime incontinence.

The definitions of continence varied among the included studies. The most common definition of daytime continence was the use of ≤1 pads (during the day), while for nighttime continence, 0–1 protection pads per night were used to define continence. According to these definitions, the 12-month rates of daytime and nighttime continence (84.2% (95% CI: 78.7–89.1%) versus 61.7% (95% CI: 51.9–71.1%) demonstrate that although the rates of daytime continence of the ONB seem satisfactory, the nighttime incontinence rates are still high and rather unacceptable. The heterogeneity of the reported data (as well as the definition and timing of continence evaluation, use of bladder diaries and questionnaires) hindered the evaluation of the chronological change of the continence outcomes of the ileal ONBs.

Furthermore, only 23 studies reported the rate of the 12 mo CIC use [8,10,13,15,17,18,20,21,22,23,24,25,26,27,28,29,30,31,32,33,34,35,36,37,39,40,48,50,52,57,60,64,65] while the number of CIC/24 h was specified only in 3 [13,37,52]. This is another important shortcoming, considering that emptying failure is common in patients with ONB [72].

Further efforts should be undertaken to identify, for each case of UD, the right test for its functional assessment based on safety, results, and cost-effectiveness. Non-invasive urodynamics as well as the use of standardised clinical evaluation, uroflowmetry, and ultrasound may be useful in this setting, limiting the use of invasive and expensive tests only to selected cases.

In our opinion, the aspects that should be standardised in ONB functional outcomes and UDS evaluation are: (1) a detailed strategy for the postoperative care of the ONB; (2) standardized reporting of the functional outcomes (definition of continence, timing of evaluation, questionnaires to be adopted etc); (3) a standardized method of conduction of the UDS (e.g., velocity of bladder filling, determination of neobladder capacity, evaluation of ONB compliance, study of the ONB voiding etc.), the proper terminology to use (e.g., how to define a contraction of the ONB during the filling phase?); (4) timing for UDS evaluation after surgery; (5) indication and reporting of the use of CIC.

## 5. Conclusions

The orthotopic neobladder has become the preferred UD after radical cystectomy in men and women. Among the various options for UD, the ONB most closely resembles the original bladder (both in location and function), and it is associated to a better quality of life for the patients. However, many ONB patients develop voiding dysfunction and other lower urinary tract symptoms, but the literature about functional outcomes is scarce.

This systematic review and meta-analysis shows a great heterogeneity of data and a total lack of standardisation in reporting functional outcomes (definition of continence, timing of evaluation, questionnaires to be adopted, UDS terminology and method).

Considering the importance of functional aspects for patients with neobladder, it is necessary to establish UDS guidelines for the setting of ONB. To this purpose, adequately designed prospective trials adopting standardised postoperative care, terminology, and methods of proper conduct of the UDS and outcome evaluation are necessary.

## Figures and Tables

**Figure 1 cancers-16-01253-f001:**
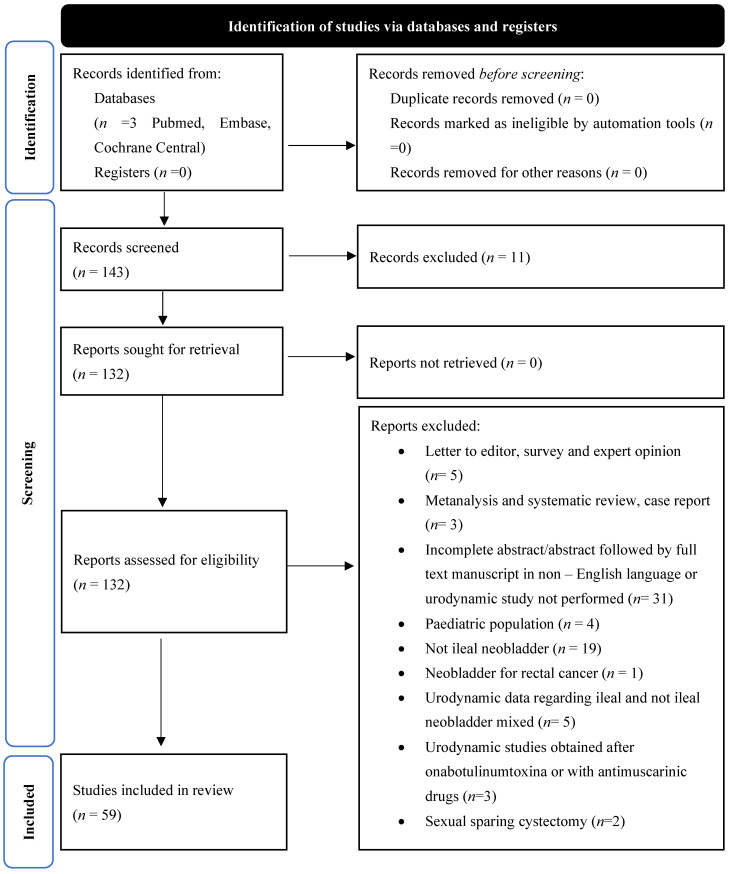
Flow of information through the different phases of this systematic review according to the PRISMA criteria.

**Figure 2 cancers-16-01253-f002:**
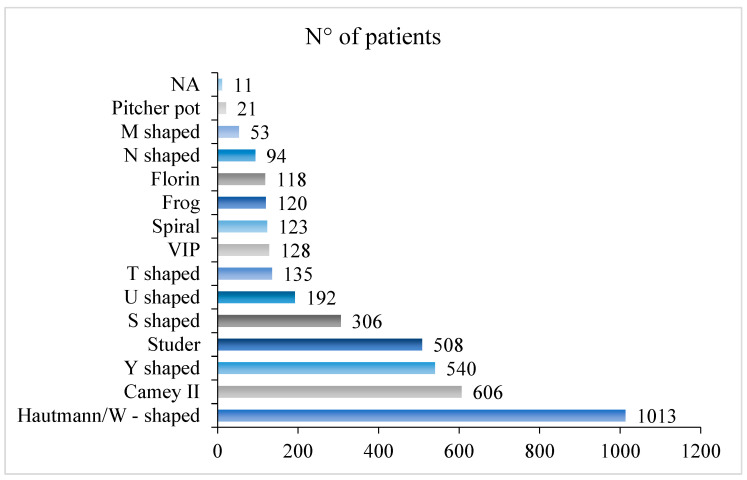
The distribution of the ileal neobladders of the manuscripts included in the present SR.

**Figure 3 cancers-16-01253-f003:**
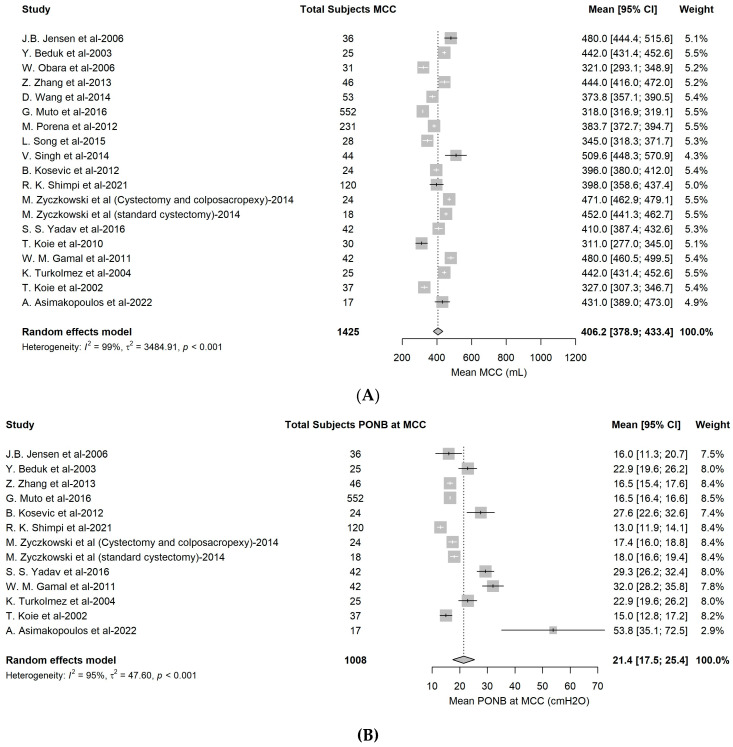
(**A**) Forest plots of mean MCC. (**B**) Forest plots of P_ONB_ at mean MCC. (**C**,**D**) Forest plots of daytime and night-time continence rates at 12 months.

**Figure 4 cancers-16-01253-f004:**
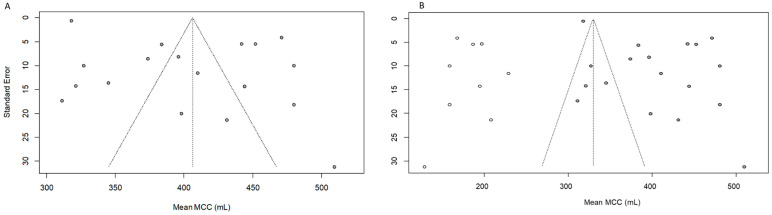
(**A**) Funnel plots of the assessment of potential publication bias in meta-analysis of MCC at 12 months and (**B**) funnel plot post trim and fill analysis. Filled circles: observed findings; open circles: imputed and added studies after trim and fill analysis.

**Figure 5 cancers-16-01253-f005:**
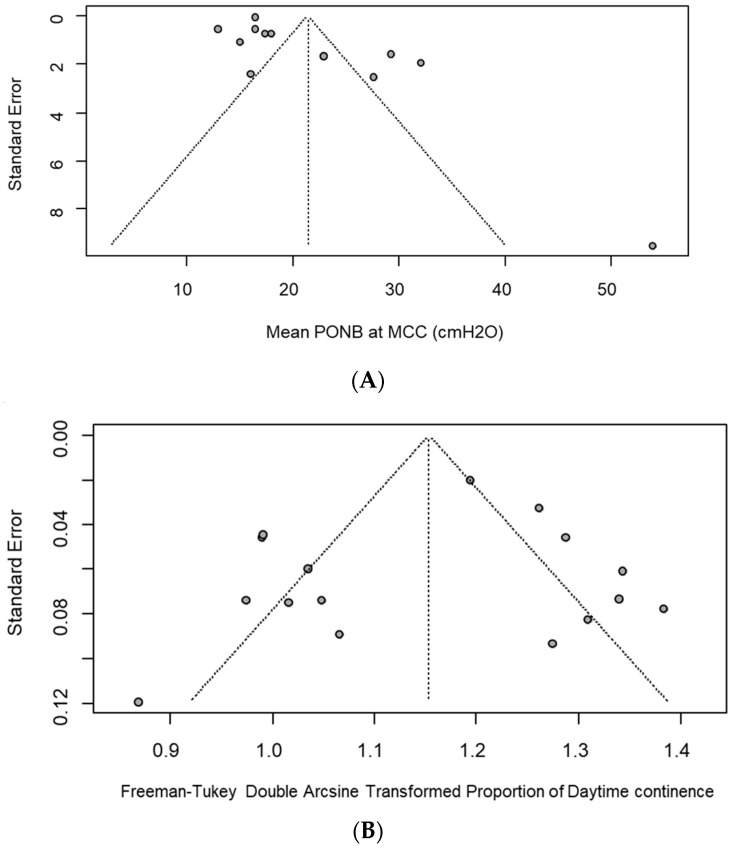
Funnel plots of the assessment of potential publication bias in meta-analysis of (**A**) P_ONB_ at MCC, (**B**) daytime and (**C**) night-time continence.

**Table 1 cancers-16-01253-t001:** Baseline characteristics used for data extraction, functional and UDS parameters collected and reported in database.

Baseline Characteristics
▪ Age (years) ▪ Sex (M vs. F vs. both) ▪ Comorbidities ▪ Type of surgical access (open vs. laparoscopic vs. robotic) ▪ Total operative time ▪ Time for ONB construction ▪ Length of ileum ▪ Type of ONB ▪ Use of stitches or mechanical stapler for configuration reservoir ▪ Creation of afferent limb ▪ Timing of urethral anastomosis (before vs. after the creation of posterior plate of ONB) ▪ Type of uretero—neobladder anastomosis (end-to-end vs. side-to-side) ▪ Antireflux ureteral anastomosis ▪ Presence of contralateral crossing of the (left) ureter ▪ Method of stenting of the implanted ureter (transurethral vs. percutaneous) ▪ Performance of nerve sparing surgery
**Functional parameters**
▪ Definition and rate of daytime and night—time continence ▪ Use of the pads ▪ Need for clean intermittent catheterization (CIC) ▪ Timing of UDS ▪ Numbers of voids day/night ▪ Score of validare questionnaires on QoL
**Free uroflowmetry**
▪ Voided volume (mL) ▪ Peak flow rate (mL/s) ▪ Postvoid residual (mL)
**UDS parameters**
▪ Presence of non-inhibited peristaltic contractions ▪ Compliance (mL/cmH_2_O) ▪ Maximum cystometric capacity (MCC, mL) ▪ First neobladder sensation (mL) ▪ Maximum pressure of ONB (cmH_2_O) ▪ ONB pressure at MCC (P_ONB_ at MCC, cmH_2_O) ▪ Coughing leak point pressure (CLPP, cmH_2_O) ▪ Valsalva leak point pressure (VLPP, cmH_2_O) ▪ Peak flow rate (Qmax, mL/s) ▪ Pressure at Qmax (cmH_2_O)

**Table 2 cancers-16-01253-t002:** Included studies.

Authors	Years of Pubblications	Type of The Studies	Main Study Endpoints	N° of Patients	IRB Approval *	Continence Data	UDS Data **
M Apostolos et al. [7]	2015	Retrospective study	To determine accuracy of UDS in neobladder	32	No	Yes	Yes
X Zhou et al. [8]	2020	Randomized Clinical trial	Perioperative and function outcome intracorporeal ONB	40	Yes	Yes	Yes
Y N Niu et al. [9]	2010	Abstract	To evaluate function and upper tract functioning in T neobladder series	90	No	Yes	Yes
M Porena et al. [10]	2012	Retrospective longitudinal study	Long—term functional outcome on ON	237	No	Yes	Yes
J B Jensen et al. [11]	2006	Retrospective study	Complication and functional outcome of the Hautmann neobladder	67	No	Yes	Yes
U P Singh et al. [12]	2019	Prospective study	Short term voiding and urodynamic outcome of W shaped iON	41	No	Yes	Yes
R Satkunasivam et al. [13]	2015	Prospective study	Functional, QoL and bladder cancer specific features of iONB	107	Yes	Yes	Yes
H Zhong et al. [14]	2019	Retrospective study	UDS and QoL outcome in iON with orthotopic ureteral reimplantation	72	Yes	Yes	Yes
R K Shimpi et al. [15]	2021	Prospective single center study	To evaluate Frog ileal neobladder	120	Yes	Yes	Yes
D Fontana et al. [16]	2004	Retrospective study	Clinical and functional outcome of Y shaped ON	53	No	Yes	Yes
T Koie et al. [17]	2006	Retrospective study	Advantage of the Goodwin method in modified ON	95	No	Yes	No
H A El—Helaly et al. [18]	2019	Retrospective study	Clinical outcomes between sigmoid and ileal neobladder	27	No	Yes	Yes
Z Zhang et al. [19]	2013	Retrospective study	UDS of N shaped ileal neobladder for 12 months	52	No	Yes	Yes
Grobet-Jeandin E et al. [20]	2021	Retrospective observational study	Urodynamic assessment and quality of life outcomes in a rONB	14	Yes	Yes	Yes
Y R Barapatre et al. [21]	2013	Prospective study	UDS outcome of W shaped iON with serosa—lined tunnel uretero—ileal anastomosis	17	Yes	Yes	Yes
V Singh et al. [22]	2014	Prospective non randomized trial	UDS and functional outcomes in orthotopic sigmoid vs. iON	44	Yes	Yes	Yes
Checcucci E et al. [23]	2021	Prospective controlled trial	Postoperative complications and functional and UDS outcomes in a case series	90	No	Yes	Yes
G Marim et al. [24]	2007	Retrospective Follow—up study	Long term functional outcomes and UDS of W—shaped iON	20	No	Yes	Yes
S S Yadav et al. [25]	2016	Prospective study	Long term functional, urodynamic, and metabolic outcomes of neobladder	42	Yes	Yes	Yes
T Koie et al. [26]	2010	Retrospective cohort study	Oncological and voiding functional outcomes after ON	30	No	Yes	Yes
G Muto et al. [27]	2016	Retrospective study	Outcomes of large series of stapled ileal orthotopic neobladder	606	No	Yes	Yes
A Khan et al. [28]	2021	Retrospective study	Functional outcomes of intracorporeal vs. extracorporeal neobladder	40	No	Yes	Yes
G Sevin et al. [29]	2004	Retrospective study	10 years—Clinical, urodynamic, functional, radiological, and metabolic outcomes of ON	124	No	Yes	Yes
A A Hassan et al. [30]	2007	Retrospective follow—up study	Functional results of Y shaped ON with antireflux ureteral reimplantation	120	No	Yes	Yes
JK Nam et al. [31]	2013	Retrospective follow—up study	>10 years postoperatively functional outcomes and UDS in a Studer neobladder	19	No	Yes	Yes
A Minervini et al. [32]	2017	Retrospective study	UDS outcomes in robotic intracorporeal neobladder configuration	18	No	Yes	No
A E Dellis et al. [33]	2014	Prospective study	Continence and urodynamic findings after modified S ileal neobladder	208	Yes	Yes	Yes
Y Bedük et al. [34]	2003	Retrospective follow—up	Clinical and UDS in ileocecal and ileal bladder substitution	36	No	Yes	Yes
K H Kim et al. [35]	2017	Retrospective study	Voiding pattern in patients with orthotopic neobladder	142	No	Yes	No
K Nagahama et al. [36]	2002	Abstract (article in Korean)	Urodynamic and functional outcome in Hautmann ileal neobladder	19	No	Yes	No
R B dos Reis et al. [37]	2011	Retrospective study	ON reconstruction in patients with shortened mesentery	5	No	Yes	Yes
S Muto et al. [38]	2007	Retrospective case control study	Changes in neobladder configuration during real time MRI	10	No	No	Yes
M Ferriero et al. [39]	2009	Retrospective comparative study	Data of Padual ileal neobladder series	46	No	Yes	Yes
W Wang et al. [40]	2012	Retrospective follow—up study	Modified spiral orthotopic ileal neobladder	51	Yes	Yes	Yes
W M Gamal et al. [41]	2011	Retrospective study	Feasibility and outcomes of the N shaped pouch	42	No	Yes	Yes
W Obara et al. [42]	2006	Retrospective study	Feasibility of Studer ON for aged patients	31	No	Yes	Yes
S Crivellaro et al. [43]	2009	Retrospective study	Functional and anatomical differences among three ON using 3D CT and videoUDS	12	No	Yes	Yes
S Rawal et al. [44]	2006	Retrospective study	Initial results of a newly modification of Studer neobladder	21	No	Yes	Yes
Z Bayraktar et al. [45]	2001	Retrospective study	UDS of 8 female patients with iON	8	No	Yes	No
B Kosevic et al. [46]	2012	Prospective clinical trial	UDS of modified orthotopic ileal neobladder	24	No	No	Yes
C Constantinides et al. [47]	2001	Retrospective study	5-year experience in a modification of S ileal pouch	43	No	Yes	Yes
M Khafagy et al. [48]	2006	Randomized controlled trial	To compare ileocecal orthotopic bladder vs. iON	29	No	Yes	No
Z Chen et al. [49]	2009	Randomized controlled trial	Continence after creation of orthotopic ileocolonic and iON	38	No	Yes	Yes
G Palleschi et al. [50]	2015	Retrospective study	Functional outcome of laparoscopic cystectomy and intracorporeal iON	30	No	Yes	Yes
D B Fang et al. [51]	2012	Retrospective study	Functional result of W ileal neobladder by a hand—assisted- drawing—needle running suture	347	No	Yes	Yes
B P Schrier et al. [52]	2005	Retrospective study	Continence rates and UDS in ileal vs. sigmoid neobladder	62	No	Yes	Yes
D Wang et al. [53]	2014	Prospective observational	UDS after laparoscopic radical cystectomy and iON	53	Yes	No	Yes
L Song et al. [54]	2014	Abstract	Functional outcomes of iON in women	28	No	Yes	No
P Honeck et al. [55]	2009	Retrospective follow—up study	Long term outcomes of sigmoid neobladder vs. iON	10	No	Yes	Yes
A Skolarikos et al. [56]	2004	Retrospective study	Continence status and UDS in ON	55	No	Yes	Yes
T Koie et al. [57]	2002	Retrospective follow—up study	Surgical and functional outcomes using a modified Goodwin technique	37	No	Yes	Yes
N Caproni et al. [58]	2006	Retrospective follow—up study	Morphofunctional evaluation of orthotopic reservoir using TC	30	No	No	Yes
M S El—Bahnasawy et al. [59]	2005	Retrospective study	UDS in patients with detubularized urinary diversion with enuresis	25	No	No	Yes
M Zyczkowski et al. [60]	2015	Randomized Clinical trial open label	Functional result in surgical modification iON	42	No	No	Yes
S Y Wang et al. [61]	2012	Retrospective Follow—up study	Outcome of laparoscopic radical cystectomy	11	No	No	Yes
K Türkölmez et al. [62]	2004	Retrospective study	Outcomes in W—shaped ON using ureteral anastomosis—serous lined extramural tunnel	42	No	Yes	No
Y Osman et al. [63]	2004	Prospective controlled trial	Long term outcomes in two reflux prevention technique in ileal neobladder	30	No	Yes	No
A Asimakopoulos et al. [64]	2022	Prospective clinical study	Urodynamic features and continence of the iYNB and (HRQoL) outcomes	26	Yes	Yes	Yes
Di Maida F et al. [65]	2022	Prospective controlled trial	Functional and urodynamic features of Florin neobladder vs. VIP	158	Yes	Yes	Yes

* IRB = Institutional Review Board. ** Complete urodynamic data: reported data on filling and voiding phase.

**Table 3 cancers-16-01253-t003:** Technical issues as described in the included manuscripts.

Technical Issues		N° (%)
Ureteral—neobladder anastomosis	End—to—end	132 (3.3)
	Side—to—side	3620 (91.2)
	NA	215 (5.4)
Antireflux ureter anastomosis	Yes	1381 (34.8)
	No	1322 (33.3)
	NA	1265 (31.9)
Crossing of the left ureter	Yes	999 (25.2)
	No	2821 (71.0)
	NA	148 (3.7)
Stenting of reimplanted ureter	Transabdominal	505 (12.7)
	Transurethral	1633 (41.2)
	NA	1830 (46.1)
Pouch configuration	Suture	1166 (29.4)
	Stapler	1175 (29.6)
	NA	1627 (41.0)
Urethro—neobladder anastomosis	At the start of reconstruction	137 (54.4)
	After reconstruction	1382 (34.8)
	NA	2449 (61.7)

## Data Availability

The raw data supporting the conclusions of this article will be made available by the authors on request.

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
