# Peer review of "Urodynamic Parameters and Continence Outcomes in Asymptomatic Patients with Ileal Orthotopic Neobladder: A Systematic Review and Metanalysis"

_cancers, 2024, doi:10.3390/cancers16071253_

Round 1

Reviewer 1 Report

Comments and Suggestions for Authors

This manuscript aims to give the first systemically review of the urodynamic outcomes of the ileal orthotopic neobladders. Despite this field received little attention that only 143 records were screened and only 59 manuscripts were eligibly included over a span of 21 years and the enrolled data were highly heterogenous, it should still be informative to confer many valuable urodynamic parameters for ileal orthotopic neobladders reconstruction. As a systemic review, it is inevitable to be a very biostatistical manuscript, but described with more medical terminology should be really helpful for urologists to realize and use those data. This review should be acceptable but most figures and tables were blurred and should be revised before publication.

Comments on the Quality of English Language

Some typographic mistakes still existed and should be revised before publication

Author Response

  • Comments and Suggestions for Authors

This manuscript aims to give the first systemically review of the urodynamic outcomes of the ileal orthotopic neobladders. Despite this field received little attention that only 143 records were screened and only 59 manuscripts were eligibly included over a span of 21 years and the enrolled data were highly heterogenous, it should still be informative to confer many valuable urodynamic parameters for ileal orthotopic neobladders reconstruction. As a systemic review, it is inevitable to be a very biostatistical manuscript, but described with more medical terminology should be really helpful for urologists to realize and use those data. This review should be acceptable but most figures and tables were blurred and should be revised before publication.

We thank the reviewer for his/her comments. This is the first ever systematic review on the urodynamic outcomes of the “normal” orthotopic neobladders (i.e for patients without symptoms). It summarizes very important outcomes (mainly urodynamic but also functional ones) and it may represent the basis (reference manuscript) for outcome comparison for other papers. It also describes the methodology of the urodynamic evaluation of the neobladders, highlighting the frequent lack of clear indications, accurate guidelines (the same parameters used for the native bladder are also used for the ileal neobladders), the absence of standardised definitions and the lack of standard values for outcome comparison. By underlining these gaps we hope that future studies will have more adequate designs and will allow for a more accurate evaluation of these patients. The manuscript may be also widely used for comparisons of the urodynamic outcomes by future manuscripts of the same context.

Comments on the Quality of English Language

Some typographic mistakes still existed and should be revised before publication

A linguistic revision was performed. We are keen to correct other mistakes (if any) at the discretion of the reviewer.

Reviewer 2 Report

Comments and Suggestions for Authors

The authors attempted in performing a study that providing the first systematic review (SR) on the urodynamic (UDS) outcomes of the ileal orthotopic neobladders (ONB). they did not make a good plan about what the real aim is, what parameters should be extracted. The results are very different from their objective. Therefore, I suggest this article is not suitable for publication in its current form.

Question 1:  There is no registered protocol for this review. The search strategy is not clearly explained.

Question 2:  Does the author cite that he used relevant articles after relevant research? Don't explain what these works are, ie what would be relevant articles / research? Are these randomized controlled trials, cohort studies, comparative studies?

In the inclusion / exclusion criteria was any language criteria used, were papers written in any language accepted?  

Question 3: In the section of the results, the subtitles should highlight the central idea of the paragraph. Therefore, the results section should be elaborated in more detail.

Question 4: It is unacceptable that the author made a great deal of discussion that they did not present in the results. 

Question 5: Again, language is a big issue for this manuscript, despite the revision. I suggest language editing service from a native speaker in the biomedical research field. 

Comments on the Quality of English Language

Extensive editing of English language required.

Author Response

  • The authors attempted in performing a study that providing the first systematic review (SR) on the urodynamic (UDS) outcomes of the ileal orthotopic neobladders (ONB). They did not make a good plan about what the real aim is, what parameters should be extracted. The results are very different from their objective. Therefore, I suggest this article is not suitable for publication in its current form.

 We thank the reviewer for his/her comments. This is the first ever systematic review on the urodynamic outcomes of the “normal” orthotopic neobladders (i.e for patients without symptoms). It summarizes very important outcomes (mainly urodynamic but also functional ones) and it may represent the basis (reference manuscript) for outcome comparison for other papers. It also describes the methodology of the urodynamic evaluation of the neobladders, highlighting the frequent lack of clear indications, accurate guidelines (the same parameters used for the native bladder are also used for the ileal neobladders), the absence of standardised definitions and the lack of standard values for outcome comparison. By underlining these gaps we hope that future studies will have more adequate designs and will allow for a more accurate evaluation of these patients. The manuscript may be also widely used for comparisons of the urodynamic outcomes by future manuscripts of the same context.

Question 1:  There is no registered protocol for this review. The search strategy is not clearly explained.

The search strategy has been re-written. We hope that in its current form it is clear for the reader. PROSPERO registration has not been performed; however, in the journal guidelines for systematic reviews the registration, although encouraged, is not mandatory and the registration information should be provided only if available.

Question 2:  Does the author cite that he used relevant articles after relevant research? Don't explain what these works are, ie what would be relevant articles / research? Are these randomized controlled trials, cohort studies, comparative studies?

The information sources, selection process and the overall search strategy has been described in the lines 77-81. The reasons for manuscript exclusion are described in lines 86-96. Lines 229-231 finally describe the quality of the included manuscripts.

In the inclusion / exclusion criteria was any language criteria used, were papers written in any language accepted? 

We added the lines 91-92, specifying that only English manuscripts have been included.

 Question 3: In the section of the results, the subtitles should highlight the central idea of the paragraph. Therefore, the results section should be elaborated in more detail.

We re-elaborated all the subsections of the results. We hope that now the data is presented in a more concise and clear way.

Question 4: It is unacceptable that the author made a great deal of discussion that they did not present in the results.

Discussion has been revised. We believe that it reflects the main issues presented the Results and in the various tables and figures. We are keen to provide more specific modifications if requested by the reviewer.

Question 5: Again, language is a big issue for this manuscript, despite the revision. I suggest language editing service from a native speaker in the biomedical research field.

A further linguistic revision of the manuscript has been performed. We are keen to provide more specific modifications if requested by the reviewer.

Reviewer 3 Report

Comments and Suggestions for Authors

Authors provided a systematic review entitled “URODYNAMIC PARAMETERS AND CONTINENCE OUT-COMES IN ASYMPTOMATIC PATIENTS WITH ILEAL OR-THOTOPIC NEOBLADDER: A SYSTEMATIC REVIEW AND METANALYSIS”. I would suggest authors changing something in the title in order to be more attractive and challenging; an example could be: “Unveiling the Nexus: Urodynamic Mastery and Continence Triumphs in Asymptomatic Individuals with Ileal Orthotopic Neobladder – A Pinnacle of Precision through Systematic Review and Metanalysis”

Issue Nr. 1

The introduction provides a quite small overview of the historical context and significance of orthotopic neobladder (ONB) as a urinary diversion method. However, some elements could be added or expanded upon to enhance the introduction:

Research Gap and Importance: Clearly state the gap in current knowledge or understanding that your research aims to address. Emphasize the importance of filling this gap to advance the field of urology and improve patient outcomes.

Specific Research Questions/Objectives: Explicitly outline the specific research questions or objectives that your systematic review aims to answer. This will provide readers with a clear understanding of the focus and scope of your study.

Rationale for Urodynamic Studies: Elaborate on why urodynamic studies (UDS) are crucial for assessing the functional aspects of ONB. Discuss the limitations of existing literature that predominantly focuses on surgical and oncological outcomes, neglecting functional aspects, and emphasize the need for a more comprehensive understanding of ONB performance.

Challenges in UDS for Ileal Neobladder: Highlight the challenges and limitations associated with applying the same UDS parameters for ONB as for an intact bladder. Discuss the anatomical and physiological differences and why a tailored approach is necessary.

Lack of Consensus and Evidence: Emphasize the current lack of consensus regarding UDS assessment for ileal neobladder and the limited evidence concerning good practice criteria, terminology, and parameters. Stress the need for standardized approaches in evaluating the filling and voiding phases of ONB.

By incorporating these elements, the introduction will not only set the stage for your systematic review but also engage readers by clearly articulating the research gap, objectives, and the significance of your study in addressing these gaps.

Issue Nr. 2

Synthesis methods. Please write the formula separated from the text and in the center.

Issue Nr. 3

“To explore the possible causes of variation of results across the studies, regarding daytime/night – time continence and UDS parameters (MCC and PONB at MCC), subgroup analyses were performed considering:

1) the study design (retrospective, prospective), and

2) the type of ONB (Y-shape, S-shape, W–Hautmann, Camey II and Studer).” I would suggest writing a paragraph and not a pointed list.

Issue Nr. 4

Figure 2 is composed by one diagram and 2 tables. Please extract tables from this figure and report them as tables, according to the guidelines of this journal.

Issue Nr. 5

Figure 3 is composed of 4 different figures, that are screenshot with a very bad quality. Please improve this quality by re-writing and formatting everything as tables, not figure of tables.

Issue Nr. 6

Conclusions are very poor and need to be improved significantly.

Comments on the Quality of English Language

A quite good use of English, but some revisions are needed.

Author Response

3) Comments and Suggestions for Authors

Authors provided a systematic review entitled “URODYNAMIC PARAMETERS AND CONTINENCE OUT-COMES IN ASYMPTOMATIC PATIENTS WITH ILEAL ORTHOTOPIC NEOBLADDER: A SYSTEMATIC REVIEW AND METANALYSIS”. I would suggest authors changing something in the title in order to be more attractive and challenging; an example could be: “Unveiling the Nexus: Urodynamic Mastery and Continence Triumphs in Asymptomatic Individuals with Ileal Orthotopic Neobladder – A Pinnacle of Precision through Systematic Review and Metanalysis”

We thank the reviewer for his/her suggestion; we acknowledge that the title is not very attractive. However, we would like to keep the title as it is, since we believe that in its current form it clearly conveys the main purpose of the manuscript to the reader.

Issue Nr. 1

The introduction provides a quite small overview of the historical context and significance of orthotopic neobladder (ONB) as a urinary diversion method. However, some elements could be added or expanded upon to enhance the introduction:

Research Gap and Importance: Clearly state the gap in current knowledge or understanding that your research aims to address. Emphasize the importance of filling this gap to advance the field of urology and improve patient outcomes.

Specific Research Questions/Objectives: Explicitly outline the specific research questions or objectives that your systematic review aims to answer. This will provide readers with a clear understanding of the focus and scope of your study.

Rationale for Urodynamic Studies: Elaborate on why urodynamic studies (UDS) are crucial for assessing the functional aspects of ONB. Discuss the limitations of existing literature that predominantly focuses on surgical and oncological outcomes, neglecting functional aspects, and emphasize the need for a more comprehensive understanding of ONB performance.

Challenges in UDS for Ileal Neobladder: Highlight the challenges and limitations associated with applying the same UDS parameters for ONB as for an intact bladder. Discuss the anatomical and physiological differences and why a tailored approach is necessary.

Lack of Consensus and Evidence: Emphasize the current lack of consensus regarding UDS assessment for ileal neobladder and the limited evidence concerning good practice criteria, terminology, and parameters. Stress the need for standardized approaches in evaluating the filling and voiding phases of ONB.

By incorporating these elements, the introduction will not only set the stage for your systematic review but also engage readers by clearly articulating the research gap, objectives, and the significance of your study in addressing these gaps.

We sincerely thank the reviewer for his/her comments. The introduction has been re-elaborated.

Issue Nr. 2

Synthesis methods. Please write the formula separated from the text and in the center.

Performed.

Issue Nr. 3

“To explore the possible causes of variation of results across the studies, regarding daytime/night – time continence and UDS parameters (MCC and PONB at MCC), subgroup analyses were performed considering:

1) the study design (retrospective, prospective), : this subanalysis has been removed

2) the type of ONB (Y-shape, S-shape, W–Hautmann, Camey II and Studer).” I would suggest writing a paragraph and not a pointed list.

Performed.

Issue Nr. 4

Figure 2 is composed by one diagram and 2 tables. Please extract tables from this figure and report them as tables, according to the guidelines of this journal.

Performed.

Issue Nr. 5

Figure 3 is composed of 4 different figures, that are screenshot with a very bad quality. Please improve this quality by re-writing and formatting everything as tables, not figure of tables.

Performed.

Issue Nr. 6

Conclusions are very poor and need to be improved significantly.

Performed.

Comments on the Quality of English Language

A quite good use of English, but some revisions are needed.

Performed.

Reviewer 4 Report

Comments and Suggestions for Authors

Dear Authors, I read with interest your manuscript. The topic is interesting, also in the light of more diffusion of robotic approach.

The methodology is good.

Below my comments:

- can you register the review on PROSPERO?

- please add AMSTAR scale as supplementary

- as well the PRISMA checklist

- are there any difference in the outcomes evaluated between open/lap/rob approaches?

Author Response

4) Comments and Suggestions for Authors

Dear Authors, I read with interest your manuscript. The topic is interesting, also in the light of more diffusion of robotic approach.

The methodology is good.

We sincerely thank the reviewer for his/her comments.

Below my comments:

 - can you register the review on PROSPERO?

PROSPERO registration has not been performed and it cannot be retrospectively performed. However, in the journal guidelines for systematic reviews the registration although encouraged, is not mandatory and the registration information should be provided only if available.

- please add AMSTAR scale as supplementary

Performed.

- as well the PRISMA checklist

Prisma checklist has been added.

- are there any difference in the outcomes evaluated between open/lap/rob approaches?

We thank the reviewer for the issue raised. However, the small numerosity of the studies on each surgical approach combined to the drop-out rates do not allow for group comparisons.

Round 2

Reviewer 3 Report

Comments and Suggestions for Authors

Authors provided a revised version of their paper.

I would convert title’s capital letters into low-case letters.

Although the authors provided a revised version of their paper, there are still some issues that need to be addressed. Here is the list:

The introduction was partially but not substantially expanded.

I do not understand the Appendix 1. It seems not to be complete. Moreover, there are some formatting issues. Please have a check and consider the substitution with a full table, instead of Appendix.

In this appendix, please consider indicating 2 as subscript in definitions of unit of measure such as “cmH2O”

Pag. 5 the format of text is different than the previous and following paragraphs.

Formula at the beginning of pag 6 is not numbered and not clearly defined. This ratio should be equal to a defined variable.

Figure 1. The word defined on the left as Identification, Screening and Included are not correctly visible. Please check them out.

Appendix 2 could be transformed into a Table. Format should be different.

Focus of figure 3A is not clear. Please, improve it.

Same observation for figure 3B. perhaps these two figures could be transformed into tables.

Discussion: “300–500 ml” please convert to “mL”.

Figure 6 is not a figure but a table. However, the content of this table/figure should be absolutely be part of a simple paragraph of this section. Table is not necessary here.

Conclusions need to be expanded.

Comments on the Quality of English Language

A quite good use of English

Author Response

Comments and Suggestions for Authors

Authors provided a revised version of their paper.

I would convert title’s capital letters into low-case letters.

Performed.

Although the authors provided a revised version of their paper, there are still some issues that need to be addressed. Here is the list:

The introduction was partially but not substantially expanded.

We performed a further expansion of the introduction.

I do not understand the Appendix 1. It seems not to be complete. Moreover, there are some formatting issues. Please have a check and consider the substitution with a full table, instead of Appendix.

Performed.

In this appendix, please consider indicating 2 as subscript in definitions of unit of measure such as “cmH2O”

Performed.

Pag. 5 the format of text is different than the previous and following paragraphs.

Text corrected.

Formula at the beginning of pag 6 is not numbered and not clearly defined. This ratio should be equal to a defined variable.

Performed.

Figure 1. The word defined on the left as Identification, Screening and Included are not correctly visible. Please check them out.

Corrected.

Appendix 2 could be transformed into a Table. Format should be different.

Performed.

Focus of figure 3A is not clear. Please, improve it.

A high resolution image has been created.

Same observation for figure 3B. perhaps these two figures could be transformed into tables.

A high resolution image has been created.

Discussion: “300–500 ml” please convert to “mL”.

Performed.

Figure 6 is not a figure but a table. However, the content of this table/figure should be absolutely be part of a simple paragraph of this section. Table is not necessary here.

Figure was removed and the relative text was included in the manuscript.

Conclusions need to be expanded.

Performed.

Round 3

Reviewer 3 Report

Comments and Suggestions for Authors

Authors responded to this second revision round point by point. All the issues were addressed properly. According to my opinion, the paper now deserves to be published.